# Dynamics of Immune Cell Infiltration and Fibroblast-Derived IL-33/ST2 Axis Induction in a Mouse Model of Post-Surgical Lymphedema

**DOI:** 10.3390/ijms26031371

**Published:** 2025-02-06

**Authors:** Kazuhisa Uemura, Kei-ichi Katayama, Toshihiko Nishioka, Hikaru Watanabe, Gen Yamada, Norimitsu Inoue, Shinichi Asamura

**Affiliations:** 1Department of Plastic Surgery, Wakayama Medical University, 811-1 Kimiidera, Wakayama 641-8509, Japan; w1121m35@wakayama-med.ac.jp (K.U.); tn@wakayama-med.ac.jp (T.N.); hikawat1447s@wakayama-med.ac.jp (H.W.); genyama77@yahoo.co.jp (G.Y.); asamura@wakayama-med.ac.jp (S.A.); 2Department of Molecular Genetics, Wakayama Medical University, 811-1 Kimiidera, Wakayama 641-8509, Japan; katayama@wakayama-med.ac.jp

**Keywords:** lymphedema, CD4^+^ T cell, IL-33, fibroblast, ST2

## Abstract

Lymphedema is an intractable disease most commonly associated with lymph node dissection for cancer treatment and can lead to a decreased quality of life. Type 2 T helper (Th2) lymphocytes have been shown to be important in the progression of lymphedema. The activation of IL-33 and its receptor, the suppression of tumorigenicity 2 (ST2) signaling pathway, induces the differentiation of Th2 cells, but its involvement in lymphedema remains unclear. In the present study, we analyzed the dynamics of immune cell infiltration, including the IL-33/ST2 axis, in a mouse tail lymphedema model. Neutrophil infiltration was first detected in the lymphedema tissue on postoperative day (POD) 2. Macrophage infiltration increased from POD 2 to 5. The number of CD4^+^ T cells, including 50% Tregs, gradually increased from POD 14. The mRNA expression of *ll13* and *Ifng* increased on POD 21. The expression of IL-33 was induced in fibroblast nuclei within dermal and subcutaneous tissues from POD 2, and the expression of the *Il1rl1* gene encoding ST2 increased from POD 7. We demonstrated the infiltration process from innate to acquired immune cells through the development of a mouse tail lymphedema. The IL-33/ST2 axis was found to be induced during the transition from innate to acquired immunity.

## 1. Introduction

Lymphedema is defined as the accumulation of lymphatic fluid and fat in the interstitial space of limb owing to the dysfunction of the lymphatic transport system, which collects and returns lymphatic fluid to the bloodstream [1,2]. Physical complaints and pain reduce the quality of life of patients with lymphedema [3,4,5]. Conservative and surgical treatments for lymphedema are used for preventing disease progression and improving symptoms [6,7,8,9], with currently incurable outcomes [10,11].

The initial lymphatic damage leads to protein-rich fluid accumulation, which promotes inflammation and fibrosis. If the collateral lymphatic vessels fail to compensate, these changes exacerbate fluid accumulation and lead to a vicious cycle of lymphatic dysfunction [12,13]. Chronic lymphedema causes the infiltration of various inflammatory cells, including neutrophils, macrophages, dendritic cells, and lymphocytes, into the lymphedema tissue [13,14,15]. In a mouse tail lymphedema model, CD4-deficient mice have been reported to exhibit significantly reduced lymphedema development compared to wild-type mice [16]. Furthermore, the inhibition of Type 2 T helper (Th2) cytokines by blocking interleukin (IL)-4 or IL-13 has been found to prevent the initiation and progression of lymphedema [16]. In addition, regulatory T cells (Tregs), among the CD4^+^ T cells, infiltrate lymphoedema tissues and suppress the Th2 immune response and tissue fibrosis, thereby improving lymphatic function [17]. However, the timing of immune cell infiltration into lymphedema tissues and the process of induction of the acquired immune system, including Th2 cell infiltration, remain unclear.

Recently, IL-33 has been reported to be involved in Th2 cell differentiation [18]. IL-33 is activated in various diseases, including asthma, atopic dermatitis, and epidural fibrosis [19,20]. It has also been shown to be involved in various biological processes, such as wound healing, systemic homeostasis, tissue repair, and regeneration [13,14]. IL-33 is mainly localized in the nuclei of epithelial and endothelial cells. It is released from the nucleus upon necrosis, binds to its receptor, suppression of tumorigenicity 2 (ST2), and induces interactions with the signaling molecule IL-1 receptor accessory protein [21]. ST2 receptors are expressed on various immune cells, including myeloid cells and lymphocytes [22]. Group 2 innate lymphoid cells (ILC2s) are known as one of the cells that express the ST2 receptors and are involved in Th2 cell differentiation. ILC2s are activated by the epithelial cell-derived cytokines IL-33, IL-25, and thymic stromal lymphopoietin [23] and potentiate Th2 cell responses via antigen presentation on the major histocompatibility complex class II [24]. ILC2 themselves also produce Th2 cytokines such as IL-5 and IL-13 [19,25]. As mentioned previously, IL-33 signaling via ST2 receptors (the IL-33/ST2 axis) has been proposed to be involved in a variety of Th2 cell-mediated diseases and may be involved in lymphedema.

In the present study, we analyzed the dynamics of immune cell infiltration, including the IL-33/ST2 axis, in a mouse tail lymphedema model.

## 2. Results

### 2.1. Early Postoperative Edema Development in a Mouse Tail Lymphedema Model

Mouse tail models are widely used in lymphedema research [14,16,26,27,28]. This edema symptom is generally observed three weeks postoperatively to analyze chronic lymphedema [14,16]. However, to understand the dynamics of CD4^+^ cell infiltration, it is necessary to analyze immune responses in the early stages of lymphedema development. Therefore, we first induced lymphedema and measured the dermal and subcutaneous tissue thicknesses and fibrosis to determine the process of edema development. We observed that postoperative edema began on postoperative day (POD) 2, and the tail thickness increased over time until POD 42. In addition, Masson’s trichrome staining clearly showed an increase in dermal and subcutaneous tissue thickness from POD 2, with progressive fibrosis over time (Figure 1 and Appendix A). Thus, edema was found to develop at an early stage of the disease in the current mouse tail model of lymphedema.

### 2.2. Variation in Acquired Immune Cells in Lymphedema Development

CD4^+^ T cell infiltration is considered essential for the progression of lymphedema [26]. Since thickening of the dermal and subcutaneous tissues was observed in the early stages of lymphedema, we immunohistochemically observed the infiltration of CD4^+^ T cells into the lymphedema tissue. CD4^+^ T cell infiltration began to increase gradually from POD 14, and a sustained increase was observed until POD 42 (Figure 2A,B and Appendix A). To clarify the spatial relationship between CD4^+^ T cells and lymphatic vessels during the development of lymphedema, we measured the distance between CD4^+^ T cells and lymphatic vessels on PODs 14, 21, and 42. CD4^+^ T cells accumulated significantly closer to lymphatic vessels at POD 21 (median = 76 μm) and at POD 42 (median = 61 μm) compared to POD 14 (median = 105 μm) (Figure 2C,D and Appendix A).

Furthermore, as Tregs are known to increase during the development of lymphedema [17], we examined the infiltration of CD4^+^Foxp3^+^ Tregs into lymphedema tissues at each time point. The number of Tregs increased gradually, as did the total number of CD4^+^ T cells, and the percentage of Tregs among CD4^+^ T cells was approximately 50% throughout the period observed after POD 7 (Appendix A).

Subsequently, we examined the onset of other acquired immune cell infiltrations in the lymphedema tissue and found that the number of CD8^+^ T and B220^+^ B cells remained at relatively low levels from PODs 2 to 21, although they increased on POD 42 (Appendix A). These results indicate that CD8^+^ T and B220^+^ B cells are activated in the late stages of lymphedema, whereas CD4^+^ T cells, which are important for lymphedema progression, infiltrate the lymphedema tissue earlier than other acquired immune cell activation.

The infiltration of CD11c^+^ dendritic cells, which are antigen-presenting cells for CD4^+^ T cells, was also examined. CD11c^+^ dendritic cells were present in low numbers from PODs 2 to 14 and increased in number on POD 21 (Figure 3A,B and Appendix A). This increase lagged behind the increase in CD4^+^ T cells, suggesting the involvement of pathways other than dendritic cells that activate CD4^+^ T cells.

### 2.3. Cytokine Variation in Lymphedema Development

Since the variation in CD4^+^ T cells, which play a central role in the progression of lymphedema, was elucidated, we next focused on Th2 and Th1 cells, which are subsets of CD4^+^ T cells studied in previous lymphedema reports [17], and analyzed the variation in cytokine expression (IL-4 and IL-13 for Th2 cytokine and Ifng for Th1 cytokine) using real-time reverse transcriptase-quantitative polymerase chain reaction (RT-qPCR). The mRNA expression levels of both Th2 cytokine IL-13 and Th1 cytokine Ifng rapidly increased on POD 21, indicating the induction of a mixed Th1/Th2 immune response (Figure 4A,B). Furthermore, the expression levels of Csf2, an essential inducer of dendritic cells and macrophages for antigen presentation to CD4^+^ T cells [29,30], were analyzed, and a gradual increase was observed from PODs 2 to 21 (Figure 4C). However, the level of the Th2 cytokine IL-4 was below the detection sensitivity in the non-operated controls and at PODs 2, 7, and 21.

### 2.4. Innate Immune Cell Variation in Lymphedema Development

Chronic lymphedema induces mixed inflammatory responses involving various inflammatory cells [14]. Following the determination of the variation in acquired immune cells, we investigated the variation in innate immune cells. We observed variations in Ly6G^+^ neutrophil and F4/80^+^ macrophage infiltration in the lymphedema tissues using fluorescence immunohistochemistry. The number of neutrophils reached its first peak on POD 2 and then decreased until POD 14 but showed a tendency to increase again on POD 21 (Figure 5A,B and Appendix A). In contrast, the number of macrophages increased from PODs 2 to 5 and was constantly detected until POD 21 (Figure 5C,D and Appendix A). Thus, these cells may play an important role in the early stages of the inflammatory response in lymphedema.

### 2.5. The IL-33/ST2 Axis in Lymphedema Development

The increase in the number of CD4^+^ T cells began on POD 14, whereas the increase in the number of dendritic cells was observed on POD 21. This suggests that factors other than dendritic cells activate CD4^+^ T cells. Therefore, we focused on the IL-33/ST2 axis as a candidate mechanism for lymphedema development. We investigated the induction of the IL-33/ST2 axis during lymphedema development using fluorescence immunohistochemistry and RT-qPCR. Postoperatively, IL-33^+^ cells significantly increased on POD 2 in the dermal and subcutaneous tissues. Furthermore, the number of IL-33^+^ cells decreased on POD 7 and remained stable until POD 42 (Figure 6A,B and Appendix A). Next, the mRNA expression levels of IL-33 and its receptor, Il1rl1, which encodes the ST2 receptor, were examined. IL-33 expression also showed an early increase on POD 2 (Figure 6C), followed by an increase in Il1rl1 expression on POD 7 (Figure 6D). Membrane-bound ST2 is a functional component of IL-33 signaling, and an increase in Il1rl1 mRNA suggests that ST2^+^ cells infiltrate lymphedema tissues. The expression of IL-33 and Il1rl1 reduced on POD 21 when the number of CD4^+^ T cells prominently increased. In human lymphedema patients, it has been reported that the expression of IL-33 increased in the superficial layer of the epidermis and plays an important role in the progression of lymphedema [31]. In contrast, in the mouse tail lymphedema model, IL-33 was constitutively expressed in epidermal basal cells before lymphedema surgery, and the relative mRNA expression of IL-33 and Il1rl1 decreased on POD 21 (Figure 6C,D and Appendix A). These results suggest that the activation of the IL-33/ST2 axis in dermal and subcutaneous tissues is induced during the transition from innate to acquired immunity during lymphedema development.

### 2.6. Expression of IL-33 in Fibroblasts Within Dermal and Subcutaneous Tissues

To clarify which cells express IL-33 in the dermal and subcutaneous tissues, we examined the colocalization of IL-33 and several cell markers on POD 2. IL-33^+^ cells on POD 2 did not co-localize with the hematopoietic cell marker CD45 (Figure 7A–C). Subsequently, to examine whether IL-33 was expressed in fibroblasts, we stained for IL-33 along with several fibroblast markers, namely α-smooth muscle actin (αSMA), vimentin, heat shock protein 47 (HSP47), and S100A4. IL-33 was not expressed in αSMA-positive myofibroblasts (Figure 7D–F), but all IL-33^+^ cells on POD 2 were expressed in Vimentin^+^ (Figure 7G–I, Appendix A), HSP47^+^ (Figure 7J–L and Appendix A), or S100A4^+^ fibroblasts (Figure 7M–O and Appendix A). These results suggest that IL-33 derived from fibroblasts in dermal and subcutaneous tissues plays a role in the early stages of lymphedema development.

## 3. Discussion

This study revealed the dynamics of each immune cell type and associated cytokine changes during lymphedema progression (Appendix A). Neutrophil infiltration in the lymphedema tissue was first detected on POD 2, followed by an increase in the number of macrophages from PODs 2 to 5. Neutrophils and macrophages induce the initial inflammatory response. IL-33 was induced in fibroblast nuclei within the dermal and subcutaneous tissues from PODs 2 to 42. In response to IL-33, the expression of *Il1rl1* mRNA increased from POD 7, indicating that ST2-expressing cells infiltrated the lymphedema tissues. On POD 14, CD4^+^ T cell counts increased, suggesting a shift from innate to acquired immunity. Approximately 50% of these CD4^+^ T cells were Tregs. These Tregs are thought to suppress the further progression of lymphedema. To determine the Th1/Th2 balance in the increased CD4^+^ cells, we measured the expression of the Th1 cytokine *Ifng* and the Th2 cytokines *Il-4* and *Il-13*; however, we could not detect a shift in the Th1/Th2 balance. The expression of *Il-4*, a typical Th2 cytokine, was not observed, for reasons unknown. CD4^+^ T cells primarily increased closer to the lymphatic vessels on POD 21 and POD 42 compared with POD 14. One possible reason for this is that the dilation of lymphatic vessels may have reduced the distance between CD4^+^ T cells and the lymphatic vessels. Additionally, at the late stages of lymphedema development, the activation of lymphatic endothelial cells through lymphatic injury may induce chemokine and cytokine production, such as CCL21, to promote the infiltration and proliferation of CD4^+^ T cells around the lymphatic vessels [32,33,34]. After an increase in dendritic cells on PODs 21 and 42, along with a reduced infiltration of macrophages and neutrophils, the acquired immune cells (including CD4^+^ T, CD8^+^ T, and B220^+^ B cells) were enhanced.

An increase in neutrophil count was observed in the early stages of lymphedema development. Neutrophils reportedly infiltrate damaged tissue during the early stages of the wound-healing process [14,35], and it has been hypothesized that lymphatic effusion and tissue damage induce neutrophil infiltration [14,35]. Previous studies have shown that neutrophil infiltration was not observed in simple skin excision controls without lymphatic cauterization [36]. Our results indicate that neutrophil infiltration into the subcutaneous tissue and the nuclear induction of IL-33 in fibroblasts, but not keratinocytes, were observed in regions without skin necrosis. The results indicate that lymphatic fluid accumulation and subsequent subcutaneous tissue damage induce immune cell infiltration rather than surgical injury. Neutrophils contribute to the degradation and digestion of injured tissue and may lead to the initial inflammatory response in lymphedema. These early inflammatory responses or direct mechanical stimuli from lymphedema exacerbation may induce the expression of IL-33 in the nuclei of cells in the dermal and subcutaneous tissues [37]. IL-33 is generally expressed in epithelial barrier tissues, such as the nasal mucosal epithelium, airway epithelium, skin keratinocytes, and simple cuboidal epithelium in the stomach [38,39]. Nuclear IL-33 functions as a stored alarmin that is released when the epithelial barrier is breached [40]. This study clarified that the expression of IL-33 is induced in the nuclei of fibroblasts in dermal and subcutaneous tissues during lymphedema development but remains stable in epidermal cells. Therefore, epidermal IL-33 does not appear to contribute significantly to the development of lymphedema in this model.

IL-33 is released from fibroblasts even under mechanical stress, without cellular damage [41]. IL-33 specifically binds to ST2 and activates ST2-expressing cells such as ILC2, which secrete Th2 cytokines, including IL-5 and IL-13, leading to Th2 differentiation [18]. In this study, the expression of *Il1rl1* was induced after the increase in IL-33, suggesting that IL-33 is not only expressed in the nucleus but is also secreted from damaged fibroblasts into the lymphedema tissue, possibly leading to the increase in ST2^+^ cells. Furthermore, the expression of *Il1rl1* and *Il-13* was induced from PODs 2 to 7 and on POD 21, respectively. Therefore, since the increase in *Il1rl1* indicates the induced infiltration of ST2^+^ cells, IL-33 may be involved in the infiltration of other ST2^+^ cells besides IL-13-expressing ILC2, leading to CD4^+^ T cell and Treg cell infiltration during lymphedema development. These suggest that the IL-33/ST2 axis is involved in linking the innate and acquired immunity during the development of lymphedema. A limitation of our research is that although the IL-33/ST2 axis is induced in the early stages of lymphedema, we have not yet identified which ST2^+^ cells are activated. Previous studies have reported that IL-33 expression is induced in the epidermis of lymphedema patients [31]. In our current study, we clarified the fibroblast-derived IL-33 induction in the progression of lymphedema. Further experiments utilizing an inhibitor of the IL-33/ST2 axis and/or recombinant IL-33 will clarify the role of the IL-33/ST2 axis in lymphedema progression. Future studies should determine how the IL-33/ST2 axis affects Th2 cell infiltration and the pathological progression of lymphedema.

## 4. Materials and Methods

### 4.1. Surgical Models of Lymphedema

A mouse tail lymphedema model was created in 9–10-week-old male C56BL/6J mice (CLEA Japan, Tokyo, Japan), as described previously [28]. Mixed anesthetics (medetomidine hydrochloride [0.75 mg/kg; Medetomin; Meiji Seika Pharma, Tokyo, Japan], midazolam [4 mg/kg; Dormicum; Astellas Pharma, Tokyo, Japan], and butorphanol [5 mg/kg; Betrufal; Meiji Seika Pharma]) were administered intraperitoneally. After the mice were completely anesthetized, a 3 mm wide skin excision was made circumferentially, 1 cm distal to the base of the tail, and the lymphatic vessels in the epidermal layer were removed. After the subcutaneous injection of Patent Blue 2 cm distal to the surgical site on the mouse tail, the deep lymphatic vessels running parallel to the lateral tail vein were ligated with 10-0 nylon. After surgery, atipamezole (0.75 mg/kg; Mepatia; Meiji Seika Pharma) was intraperitoneally administered. The operated tail region was wrapped with Opsite Quick Roll (Smith & Nephew, Watford, UK) and kept moist for 24 h. The mice were euthanized via cervical dislocation to collect tissue for RNA and histological analyses. None of the mice developed tail skin necrosis following surgery.

### 4.2. Measurement of Dermal and Subcutaneous Tissue Thickness

The tail tissue was evaluated using specimens stained with Masson’s trichrome. A line was drawn horizontally from the lowest point of the muscle on the ventral surface to define the thicknesses of the dermal and subcutaneous tissues. Thickness was measured using ImageJ ver. 1.54c (Wayne Rasband, National Institutes of Health, Bethesda, MD, USA). The area and percentage of fibrotic tissue in the dermal and subcutaneous tissues were obtained from Masson’s trichrome-stained samples. Using the ImageJ software, the images were split into RGB channels, and the red channel was subtracted from the blue channel. After excluding the bone and muscle regions, the lower threshold was set at 50%, and the extracted areas were measured as fibrotic areas in the dermal and subcutaneous tissues (Appendix A).

### 4.3. Real-Time RT-qPCR Analysis

The mouse tail tissue, excluding the bone, was disrupted using a multi-bead shocker (Yasui Kikai, Osaka, Japan); total RNA was extracted using the RNeasy Mini Kit (Qiagen, Venlo, The Netherlands), and cDNA was synthesized using the SuperScript III First-Strand Synthesis System (Thermo Fisher Scientific, Waltham, MA, USA). The mouse 18 S-ribosomal gene was used as an internal control. RT-qPCR was performed using the TaqMan primer with probe sets for *Il-4*, *ll-13*, *Ifng*, *Csf2*, *Il-33* and *Il1rl1* (Mm00445259m1, Mm00434204m1, Mm01168134_m1, Mm01290062_m1, Mm00505403m1, and Mm00516117m1, respectively; Thermo Fisher Scientific). Relative gene expressions are calculated as the ratio of target gene expression at each time point for the control group or the POD 2 group with the ΔΔCT method.

### 4.4. Histology and Fluorescence IMMUNOHISTOCHEMISTRY

Tissues were harvested at an 8 mm length distal to the surgical site. Lymphoid tail tissues were fixed in 4% paraformaldehyde for 48 h at 4 °C, decalcified with 10% ethylenediaminetetraacetic acid (EDTA) in phosphate-buffered saline (PBS) for 10 days, halved on the fifth day of decalcification, and embedded in paraffin. Next, 6 μm paraffin sections were cut from the distal side of the proximal tissue and stained with Masson’s trichrome or subjected to immunohistochemistry (Appendix A). Frozen tissues were cryoprotected in 30% sucrose/PBS and embedded in an optimal cutting temperature compound (Sakura Finetek, Tokyo, Japan) with 30% sucrose/PBS. Finally, 16 μm frozen sections were cut from the proximal side of the distal tissue.

For immunofluorescence staining, antigen retrieval was performed using sodium citrate (pH 6.0) or Tris-EDTA (pH 9.0) at 121 °C for 1 min. Sections were incubated overnight at 4 °C with primary antibodies, followed by incubation with the appropriate secondary antibodies conjugated to Alexa Fluor 488 and 568 for 1 h at room temperature. The Vector TrueVIEW Autofluorescence Quenching Kit (Vector Laboratories, Newark, CA, USA) containing 4′,6-diamidino-2-phenylindole was used to reduce erythrocyte autofluorescence.

Images of the sections were obtained using a BZ X-800 microscope (Keyence, Osaka, Japan) and analyzed using ImageJ. We randomly photographed eight fields per tissue at 20× magnification and calculated the average number of cells per field. Subsequently, the average number per field for each mouse was used to derive the group mean at each time point. When cell markers or cytokines were consistently stained in nuclear-stained cells, the cells were counted manually as positive. IL-33^+^ cells were counted in dermal and subcutaneous tissues. The number of IL-33^+^ cells in the epidermis was counted within a 300 μm length in the field of view. The distances between CD4^+^ T cells and the nearest lymphatic vessels were manually measured using the ImageJ software.

### 4.5. Statistical Analysis

To compare all differences between time points, the Steel–Dwass test was used if the Shapiro–Wilk normality test showed *p* < 0.05; the Tukey test was used if the Shapiro–Wilk normality test showed *p* ≥ 0.05 and the Levene equivariance test showed *p* ≥ 0.05. The Games–Howell test was used if the Shapiro–Wilk normality test showed *p* ≥ 0.05 and the Levene equivariance test showed *p* < 0.05. Statistical significance was set at *p* < 0.05.

## 5. Conclusions

The dynamics of innate immune cells to those of acquired immune cells in a mouse tail lymphedema model were clarified. Fibroblast-derived IL-33 was found to be induced during the transition from innate to acquired immunity.

## Figures and Tables

**Figure 1 ijms-26-01371-f001:**
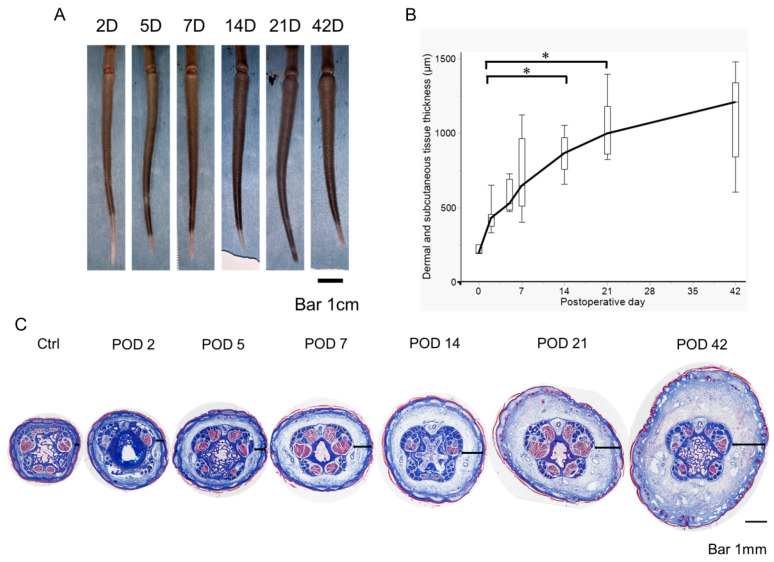
Tail dermal and subcutaneous tissue thickness in the tail lymphedema tissue in non-operated controls (Ctrl) and on postoperative days (PODs) 2, 5, 7, 14, 21, and 42. (**A**) Representative images of the tail lymphedema on different PODs. (**B**) Time course of tail thickness after surgery in C57BL/6 J mice (Ctrl: n = 3; PODs 2, 5, 7, 14, and 21: n = 8; POD 42: n = 6). Boxes represent 50% of the data, with medians (lines) and interquartile ranges (whiskers). The Steel–Dwass test was used to determine which time points were significantly different from POD 2. * *p* < 0.05. (**C**) Representative histological images of lymphedema tissues stained with Masson’s trichrome, with collagen fibers stained blue and muscle fibers stained red. Dermal and subcutaneous tissue is indicated by the double-headed arrow.

**Figure 2 ijms-26-01371-f002:**
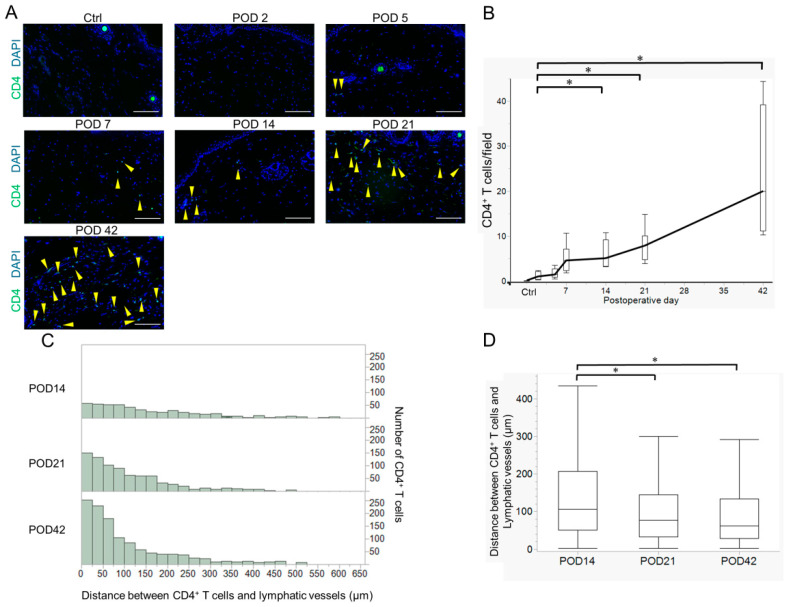
Infiltration of CD4^+^ T cells and spatial relations of CD4^+^ T cells and lymphatic vessels. (**A**) Representative images of CD4^+^ T cell (green) infiltrating the lymphedema tissues (Ctrl and PODs 2, 5, 7, 14, 21, and 42). Nuclei were stained with 4′,6-diamidino-2-phenylindole (DAPI) (blue). Yellow arrowheads indicate CD4^+^ T cells. Scale bar = 100 μm. (**B**) Variation in the numbers of CD4^+^ T cells per field (8 fields/mouse) infiltrating the lymphedema tissue (Ctrl: n = 3; PODs 2, 5, 7, 14, and 21: n = 8; POD 42: n = 6). Boxes represent 50% of the data, with medians (lines) and interquartile ranges (whiskers). The Steel–Dwass test was used to determine which time points were significantly different from POD 2. * *p* < 0.05. (**C**) The spatial relations of CD4^+^ T cells and lymphatic vessels on postoperative days (PODs) 14, 21, and 42. Histograms indicate the number of CD4^+^ T cells localized at various distances from lymphatic vessels. The distance between CD4^+^ T cells present in eight fields/mouse, and lymphatic vessels were measured at each time point (443 cells on POD 14, n = 8; 764 cells on POD 21, n = 8; 1153 cells on POD 42, n = 8). (**D**) Averages of distances between CD4^+^ T cells and lymphatic vessels at each time point. Boxes represent 50% of the data, with medians (lines) and interquartile ranges (whiskers). The Steel–Dwass test was used to determine the time points that were significantly different from POD 14. * *p* < 0.05.

**Figure 3 ijms-26-01371-f003:**
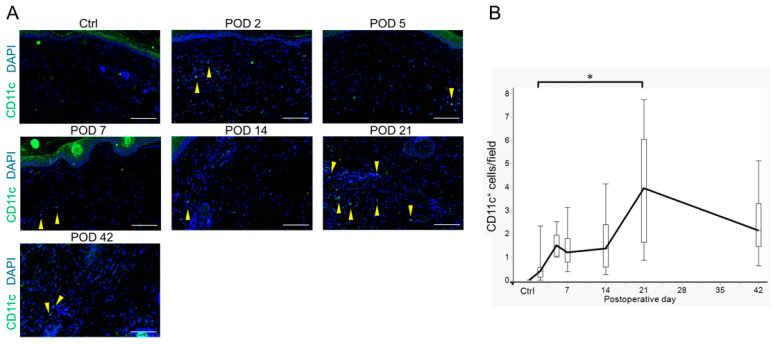
Infiltration of CD11c^+^ dendritic cells in the tail lymphedema tissue in non-operated controls (Ctrl) and on postoperative days (PODs) 2, 5, 7, 14, 21, and 42. (**A**) Representative images of CD11c^+^ dendritic cells (green) infiltrating the lymphedema tissues (Ctrl and PODs 2, 5, 7, 14, 21, and 42). Nuclei were stained with 4′,6-diamidino-2-phenylindole (DAPI) (blue). Yellow arrowheads indicate CD11c^+^ dendritic cells. Scale bar = 100 μm. (**B**) Variation in the numbers of CD11c^+^ dendritic cells per field (8 fields/mouse) infiltrating the lymphedema tissue (Ctrl: n = 3; POD 2, 5, 7, 14, and 21: n = 8; POD 42: n = 6). Boxes represent 50% of the data, with medians (lines) and interquartile ranges (whiskers). The Steel–Dwass test was used to determine the time points that were significantly different from POD 2. * *p* < 0.05.

**Figure 4 ijms-26-01371-f004:**
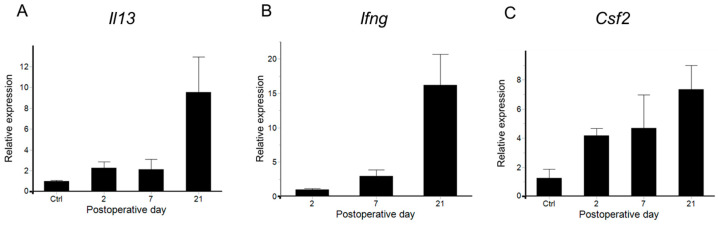
Expression of cytokines in the tail lymphedema tissue in non-operated controls (Ctrl) and on postoperative days (PODs) 2, 7, and 21. Relative expression levels of *ll-13* (**A**), *Ifng* (**B**), and *Csf2* (**C**) mRNA in tail lymphedema tissues (PODs 2,7, and 21: n = 4) were compared with those in non-operated controls (Ctrl) (n = 3). *Ifng* expression levels were below the detection sensitivity in the Ctrl group and, therefore, compared to POD 2. Relative expression levels of mRNA were normalized to 18s rRNA expression levels and were calculated with the ΔΔCT method as the ratio of the averages of expression levels of non-operated control or POD 2. Data are shown as mean ± standard error. The Games–Howell test was used for statistical analyses.

**Figure 5 ijms-26-01371-f005:**
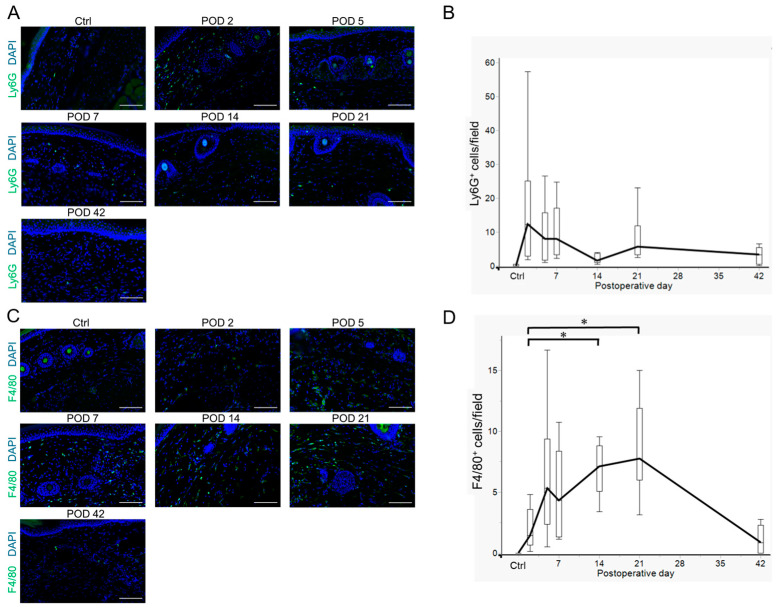
Infiltration of Ly6G^+^ neutrophils and F4/80^+^ macrophages in the tail lymphedema tissue in non-operated controls (Ctrl) and on postoperative days (PODs) 2, 5, 7, 14, 21, and 42. (**A**) Representative images of Ly6G^+^ neutrophils (green) infiltrating the lymphedema tissues (Ctrl and PODs 2, 5, 7, 14, 21, and 42). (**B**) Variation in the numbers of Ly6G^+^ neutrophils per field (8 fields/mouse) infiltrating the lymphedema tissue (Ctrl: n = 3; PODs 2, 5, 7, 14, and 21: n = 8; POD 42: n = 6). Boxes represent 50% of the data, with medians (lines) and interquartile ranges (whiskers). (**C**) Representative images of F4/80+ macrophages (green) infiltrating the lymphedema tissues (Ctrl and PODs 2, 5, 7, 14, 21, and 42). (**D**) Variation in the numbers of F4/80+ macrophages per field (8 fields/mouse) infiltrating the lymphedema tissue (Ctrl: n = 3; PODs 2, 5, 7, 14, and 21: n = 8; POD 42: n = 6). Boxes represent 50% of the data, with medians (lines) and interquartile ranges (whiskers). The Steel–Dwass test was used to determine which time points were significantly different from POD 2. * *p* < 0.05. Nuclei in (**A**,**C**) were stained with 4′,6-diamidino-2-phenylindole (DAPI) (blue). Scale bar = 100 μm.

**Figure 6 ijms-26-01371-f006:**
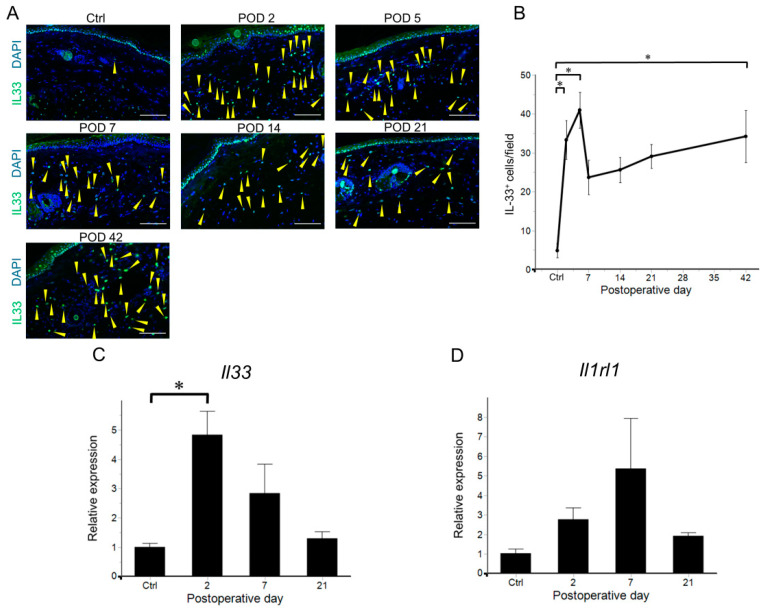
Infiltration of interleukin (IL)-33^+^ cells and express of *Il-33/Il1rl1* in the tail lymphedema tissue. (**A**) Representative images of IL-33^+^ cells (green) in the lymphedema tissues (non-operated controls [Ctrl] and postoperative days [PODs] 2, 5, 7, 14, 21, and 42). Nuclei are stained with 4′,6-diamidino-2-phenylindole (DAPI) (blue). Yellow arrowheads indicate IL-33^+^ cells in dermal and subcutaneous tissues. Scale bar = 100 μm. (**B**) Variation in the numbers of IL-33^+^ cells per field (8 fields/mouse) in the lymphedema tissue (Ctrl: n = 3; PODs 2, 5, 7, 14, and 21: n = 8; POD 42: n = 6). Data are shown as the means ± standard error. The Tukey test was used to determine which time points were significantly different from Ctrl. * *p* < 0.05. Relative expression levels of *Il-33* (**C**) and *Il1rl1* (**D**) mRNA in tail lymphedema tissues (PODs 2, 7, and 21; n = 4) were compared with those in Ctrl tissues (n = 3). Relative mRNA expression levels were normalized to 18s rRNA expression levels and calculated as the ratio of the average levels in non-operated controls. Data are shown as mean ± standard error. Tukey’s test was used to determine the time points that were significantly different from those of the Ctrl group. * *p* < 0.05.

**Figure 7 ijms-26-01371-f007:**
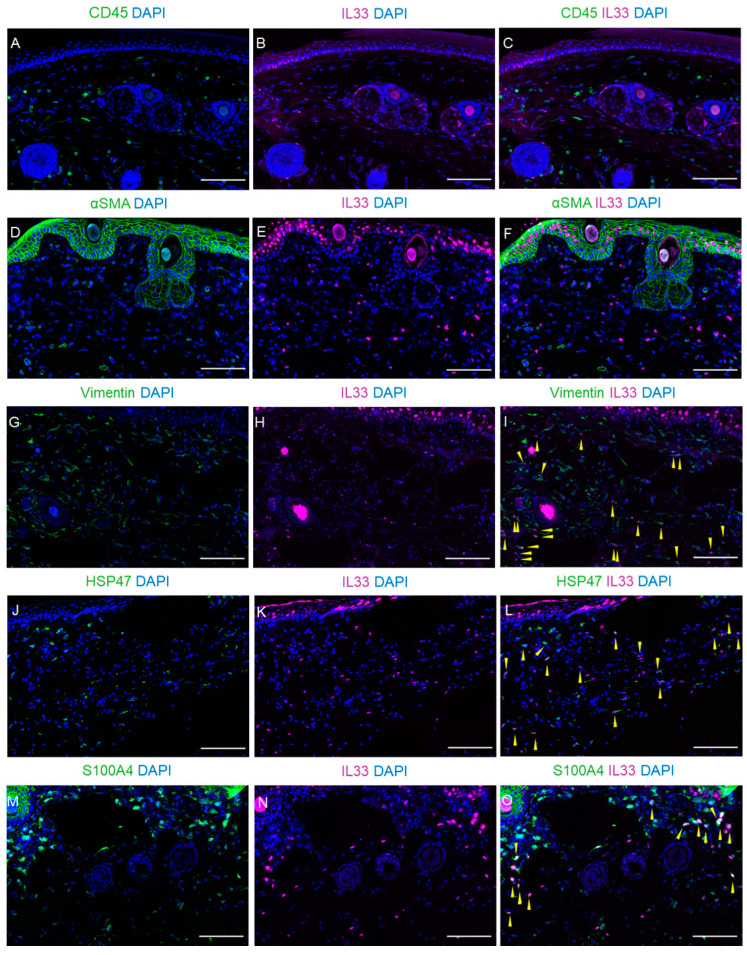
Expression of interleukin (IL)-33 in fibroblast within the lymphedema tissue on postoperative day (POD) 2. The tail lymphedema tissues on POD 2 were co-stained with IL-33 (magenta) and CD45 (**A**–**C**), α smooth muscle actin (SMA) (**D**–**F**), vimentin (**G**–**I**), heat shock protein (HSP) 47 (**J**–**L**), and S100A4 (**M**–**O**) (green). Nuclei were stained with 4′,6-diamidino-2-phenylindole (DAPI) (blue). Yellow arrowheads indicate IL-33 and fibroblast marker double-positive cells. Scale bar = 100 μm.

## Data Availability

The datasets used and/or analyzed during the current study are available from the corresponding author upon reasonable request.

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
