# Peer review of "Dynamics of Immune Cell Infiltration and Fibroblast-Derived IL-33/ST2 Axis Induction in a Mouse Model of Post-Surgical Lymphedema"

_ijms, 2025, doi:10.3390/ijms26031371_

Round 1
Reviewer 1 Report
Comments and Suggestions for Authors
In this manuscript, the authors showed dynamics of immune cell infiltration including neutrophil, macrophage, and lymphocytes in the mouse-tail lymphedema model. They considered that tissue damages induce IL-33 expression in fibroblasts and the cytokine induced ST2+ cell infiltration such as ILC2. However, the authors should revise the manuscript to explain their results and opinions more clearly.
Major Comments:
1. In Figures 6, 7, S9, and S10, I could observe IL-33 positive cells in stratum corneum even at Ctrl mouse. Additionally, the number of IL-33 positive cells at POD7 looks larger than that of POD5 in Figure 6A. The authors should indicate the cells they counted as IL-33 positive cells.
2. To introduce the contribution of IL-33/ST2 axis in lymphedema, the authors should treat the mice with an inhibitor of IL-33/ST2 axis and/or recombinant IL-33 during the lymphedema model development. Is an inhibitor of IL-33/ST2 axis able to suppress infiltration of CD4+ T cells? The authors should add the references related to IL-33/ST2 axis and lymphedema development to the Discussion.
3. The authors should explain the interpretation the changes of distance between CD4+ T cells and lymphatic vessel (Figure 2). More and more dilated lymphatic vessels were observed in lymphedema region at POD42 (Figure S3). Thus, I think the distances between CD4+ T cells and lymphatic vessels inevitably become close in lymphedema region.
Minor Comments:
1. There are several typos, please review the manuscript. For example, the strain name of mice is C57BL/6J not C57BL6/J. I could not understand means of “lymphedema tais” in lane 89.
2. Why are the authors compared the data with that of POD2 in Figures 1, 2, 3, and 5?
3. Steel-Dwass test is a non-parametric test. Thus, in my opinion, the data should be shown in median and interquartile range.
Comments on the Quality of English LanguageI can't comment on the quality of English, but there are some typos in the manuscript.
Reviewer 2 Report
Comments and Suggestions for Authors
I would like to express my sincere gratitude for the opportunity to review your manuscript. Your research is well-organized and presented in a clear and concise manner. Nevertheless, I have a few questions regarding the immunohistochemical staining. The choice of color for the staining is somewhat unclear.
I wonder if it would have been more appropriate to use a green fluorescent dye instead, as this is a more common choice in the field. Additionally, if other reviewers share this concern, I suggest that you consider converting the red color to green in the Supplementary Figure using RGB conversion.
This modification, accompanied by the original figure, would likely not have a significant impact on the overall interpretation of your results.
Round 2
Reviewer 1 Report
Comments and Suggestions for Authors
The authors revised the manuscript clearly. I have some additional question and opinion.
1. In reference #31, the expression of TSLP, IL-33, and IL-25 were increased in the superficial layer of the epidermis in lymphedema region. I agree increasing of IL-33 expressed cells in dermal and subcutaneous tissues might facilitate Th2 axis and contribute to development of lymphedema. Whereas, is the IL-33 expression in keratinocyte not contribute to development of lymphedema in the model?
2. In line 308–315, I think it is difficult to understand why “These results suggest that the IL-33/ST2 axis is involved in linking the innate and acquired immunity” from the contents of this paragraph. Is the timing of expression of IL-33 and ST2 or those mRNAs related to linking the innate and acquired immunity?
3. In line 420, please describe approval date of the animal study protocol.
Round 3
Reviewer 1 Report
Comments and Suggestions for Authors
Thank you for the re-submission of your article. I have read and confirmed the manuscript revised well.